# STRUCTEXTV2: MASKED VISUAL-TEXTUAL PREDICTION FOR DOCUMENT IMAGE PRE-TRAINING

**Yuechen Yu[†], Yulin Li[†], Chengquan Zhang[†], Xiaoqiang Zhang, Zengyuan Guo, Xiameng Qin, Kun Yao, Junyu Han, Errui Ding, Jingdong Wang**
Department of Computer Vision Technology (VIS), Baidu Inc.
`https://github.com/PaddlePaddle/VIMER/tree/main/StrucTexT`

## ABSTRACT

In this paper, we present StrucTexTv2, an effective document image pre-training framework, by performing masked visual-textual prediction. It consists of two self-supervised pre-training tasks: masked image modeling and masked language modeling, *based on text region-level image masking*. The proposed method randomly masks some image regions according to the bounding box coordinates of text words. The objectives of our pre-training tasks are reconstructing the pixels of masked image regions and the corresponding masked tokens simultaneously. Hence the pre-trained encoder can capture more textual semantics in comparison to the masked image modeling that usually predicts the masked image patches. Compared to the masked multi-modal modeling methods for document image understanding that rely on both the image and text modalities, StrucTexTv2 models image-only input and potentially deals with more application scenarios free from OCR pre-processing. Extensive experiments on mainstream benchmarks of document image understanding demonstrate the effectiveness of StrucTexTv2. It achieves competitive or even new state-of-the-art performance in various downstream tasks such as image classification, layout analysis, table structure recognition, document OCR, and information extraction under the end-to-end scenario.

## 1 INTRODUCTION

In the Document Artificial Intelligence, how to understand visually-rich document images and extract structured information from them has gradually become a popular research topic. Its main associated tasks include document image classification Harley et al. (2015), layout analysis Zhong et al. (2019), form understanding Jaume et al. (2019), document OCR (also called text spotting) Li et al. (2017); Liao et al. (2021), and end-to-end information extraction (usually composed of OCR and entity labelling phrase) Wang et al. (2021), etc. To solve these tasks well, it is necessary to fully exploit both visual and textual cues. Meanwhile, large-scale self-supervised pre-training Li et al. (2021a); Appalaraju et al. (2021); Xu et al. (2020; 2021); Huang et al. (2022); Gu et al. (2021) is a recently rising technology to enhance multi-modal knowledge learning of document images.

There are two mainstream self-supervised pre-training frameworks for document image understanding. As illustrated in Fig. 1: (a) The first category is the masked multi-modal modeling such as the proposed pre-training tasks: MLM Devlin et al. (2019), MVLM Xu et al. (2021), MM-MLM Appalaraju et al. (2021) and MSM Gu et al. (2021), whose inputs mainly consists of OCR-extracted texts and image embeddings. The methods collect semantic information from text and image, depending heavily on front-end OCR engines with certain computing costs. Additionally, the two components of OCR engine and document understanding module are separately optimized, which is hard to ensure performance of the whole system. (b) The second category is the masked image modeling (MIM) that inherits the concept of vision-based self-supervised learning such as BEiT Bao et al. (2022), SimMIM Xie et al. (2022), MAE He et al. (2022), CAE Chen et al. (2022), and DiT Li et al. (2022), etc. MIM is a powerful image-only pre-training technique to learn the visual contextualized representations of document images. Because of the insufficient consideration of textual

---

† Equal contribution. Correspondence to: Chengquan Zhang<zhangchengquan@baidu.com>.

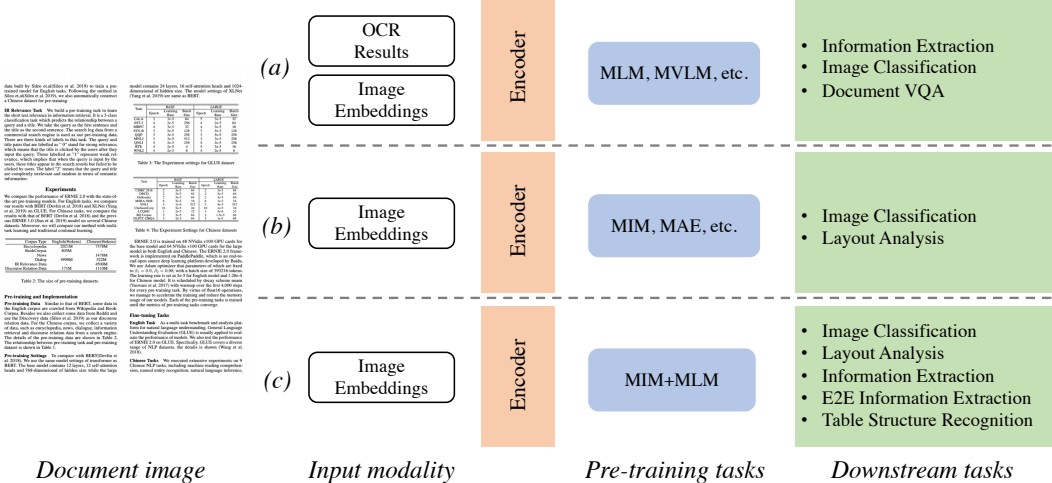

Figure 1: Comparisons with the main-stream pre-training models of document image understanding. (a) It shows the masked multi-modal modeling methods which input both OCR results and image embeddings. (b) The framework that inputs image-only embeddings is suitable for vision-dominated tasks like document image classification and layout analysis. (c) StrucTexTv2 learns visual-textual representations using only the information from images in the pre-training step and then optimizes various downstream tasks of document image understanding end-to-end.

contents, MIM often applies to some vision-dominated tasks Li et al. (2022) like image classification, layout analysis and table detection. (c) To take full advantage of multi-modality knowledge on the basis of MIM, we propose the third pre-training framework by learning visual-textual representations with image-only inputs, to optimize the performance of document image understanding tasks in an end-to-end manner.

Due to the great disparities between vision and language, the existing document understanding methods either consider a single modality or introduce an OCR engine to capture textual content in advance. Researchers used text tokens as the input in language modeling, or selected fixed-size image patches as the granularity of vision pre-training tasks. However, the textual content is visually-situated in a document and extracted from the image. Thus, we propose the text region-level image masking scheme corresponding to document content to bridge vision modeling to language modeling with the shared representations.

This paper proposes StrucTexTv2, a novel multi-modal knowledge learning framework for document image understanding by performing text region-level image masking with dual parallel self-supervised tasks of image reconstruction and language modeling (as shown in Fig. 2). First off, we adopt an off-the-shelf OCR toolkit to perform word-level text detection and text recognition on the pre-training dataset (IIT-CDIP Test Collection Lewis et al. (2006)). Next, we randomly mask some text word regions given the input images and fed them into the encoder. Finally, the pre-training objectives of StrucTexTv2 learn to reconstruct image pixels and text content of the masked words. In support of the proposed pre-training tasks, we introduce a new backbone network for StrucTexTv2. In particular, a CNN-based network with the RoI-Align He et al. (2017) operation produces visual features for the masked regions. Inspired by ViBERTGrid Lin et al. (2021), the backbone uses FPN Lin et al. (2017) to integrate features of CNN. The following transformer model enables capturing semantical and contextualized representations from the visual features. We evaluate and verify our pre-trained model in five tasks including document image classification, layout analysis, table structure recognition, document OCR, and end-to-end information extraction, all of which have achieved significant gains. The experimental results have also confirmed that the framework of StrucTexTv2 can construct fundamental pre-trained models for document image understanding.

The major contributions of our work can be summarized as following:

- A novel self-supervised pre-training framework by performing text region-level document image masking, named StrucTexTv2, is proposed to learn visual-textual representations in an end-to-end manner.

- Superior performance in five downstream tasks demonstrates the effectiveness of the Struc-TexTv2 pre-trained model in document image understanding.

## 2 RELATED WORK

**Self-supervised Learning** Thanks to the development of self-supervised tasks and Transformer architectures, in the past few years, the computer vision (CV) and natural language processing (NLP) have achieved breakthroughs in knowledge learning from large-scale unlabeled data. In the domain of NLP, Masked Language Modeling (MLM) task has been widely used in pre-trained models Devlin et al. (2019); Radford et al. (2018) due to its efficiency and effectiveness. The MLM task randomly masks a set of text tokens of input and reconstructs them according to the context around the masked tokens. In the CV domain, taking a similar idea, Masked Image Modeling (MIM) has also been successfully verified. There are several variants of MIM, for example, BEiT Bao et al. (2022) and CAE Chen et al. (2022) randomly mask a percentage of image patches and predict discrete image patch tokens learned by dVAE Ramesh et al. (2021). MAE He et al. (2022) and SimMIM Xie et al. (2022) take a simpler behavior, predicting RGB values of raw pixels by direct regression.

**Document pre-trained models** As stated above, the cutting-edge document pre-training models can be roughly separated into two categories: masked multi-modal modeling and masked image modeling approaches. The representative works of the former are LayoutLM series Xu et al. (2020; 2021); Huang et al. (2022), Docformer Appalaraju et al. (2021), UDoc Gu et al. (2021) and StrucTexT Li et al. (2021c), they usually depend on a mature OCR engine to extract textual content from document images, and fed them to the encoder combined with image embeddings. These OCR-based methods promise excellent performance, but they are cumbersome to solve some vision-dominated tasks. The latter methods, such as DiT Li et al. (2022) almost following the idea of BEiT Bao et al. (2022), directly use a general CV pre-training framework to learn on large-scale document image data. Due to the lack of guidance for text information, the model is likely to be deficient in semantic understanding. Recently, prompt-learning Liu et al. (2023) has been studied for adapting pre-trained vision-language models. Those prompt-based methods Kim et al. (2022); Davis et al. (2022) directly generate textual output from documents and achieve competitive performance on downstream tasks.

In this paper, the proposed StrucTexTv2 is a new solution that integrates the advantages of CV and NLP pre-training methods in an end-to-end manner. Benefiting from image-only input of the encoder, our framework can avoid the interference of false OCR results compared with the OCR-based pre-trained models. For our pre-training, although the supervision labels partially come from OCR results, only the high-confidence words from OCR results are randomly selected. The impacts of the OCR quality on our pre-trained model is alleviated to a certain extent.

## 3 APPROACH

### 3.1 MODEL ARCHITECTURE

As illustrated in Fig. 2, there are two main components of StrucTexTv2: a encoder network using FPN to integrate visual features and semantic features, and the pre-training framework containing two objectives: Mask Language Modeling and Mask Image Modeling.

The proposed encoder consists of a visual extractor (CNN) and a semantic module (Transformer). Given an input document image, StrucTexTv2 extracts visual-textual representations through this backbone network. Specifically, the features of the last four down-sampled stages on CNN are extracted from the visual extractor. In the semantic module, following ViT Dosovitskiy et al. (2021) to handle 2D feature maps, the features of the last stage in CNN are flattened in patch-level and are linearly projected to obtain 1D sequence of patch token embeddings, which also serves as the effective input for the Transformer. A relative position embedding representing the token index is added to the token embeddings. Then, the standard Transformer receives the input token embeddings and outputs the semantic enhanced features. We reshape the output features back to context feature maps in the 2D visual space and up-sample the feature maps with the factors of 8. We adopt FPN strategy Lin et al. (2017) to merge visual features of different resolutions from CNN and then concatenate context feature maps with them, deriving a feature map with 1/4 size of the input image.

Finally, a fusion network which consists of two successive 1×1 convolutional layers is introduced to take a further multi-modal fusion.

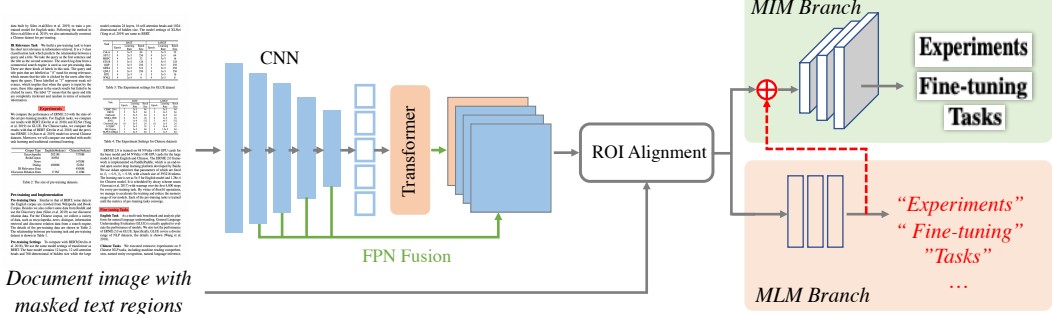

Figure 2: The overview of StrucTexTv2. Its encoder network consists of a visual extractor (CNN) and a semantic module (Transformer). Given a document image, the encoder extracts the visual feature of the whole image by CNN and obtains the semantic enhanced feature through a Transformer. Subsequently, a lightweight fusion network is utilized to generate the final representation of the image. With the help of ROI Alignment, the multi-modal feature of each masked text region is processed by the MIM branch and the MLM branch to reconstruct the pixels and text, respectively.

## 3.2 PRE-TRAINING

To enhance the document image understanding of StrucTexTv2, we perform pre-training on a large-scale document dataset IIT-CDIP Test Collection 1.0 dataset Lewis et al. (2006). Unlike MAE He et al. (2022), BEiT Bao et al. (2022), and DIT Li et al. (2022), we do not use patch-level masking strategy for pre-training. Instead, we use a novel text region-level masking strategy and employ two self-supervised pre-training tasks to learn visual-textual representations. The subsequent experiment results suggest that the fine-grained text region-level masking strategy is more suitable than the coarse-grained patch-level masking strategy in document understanding.

### 3.2.1 TASK #1: MASKED LANGUAGE MODELING

To encourage the model to learn contextual representation, similar to BERT Devlin et al. (2019), we mask a portion of the text region of the input document with RGB values [255, 255, 255] at random. We find that 30% of masking ratio is the best choice in our experiments. According to the contextual information of the unmasked regions in a document, the encoder-decoder architecture learns to predict the indexes of text tokens (in a pre-defined vocabulary) of the masked text regions with cross-entropy loss. To avoid the interference of sub-words produced by WordPiece Song et al. (2021), we only choose the first sub-word in each word for keeping the number of words in the document images. Notice that the Masked Language Modeling task in StrucTexTv2 does not require the text input, however, which is essential for the NLP domain.

*Decoder for Task #1.* For Mask Language Modeling, we employ an extremely 2-layer MLP as a decoder to project the encoding feature. The multi-model feature of each text region is extracted from the fused feature maps $\mathbf{F}_{fuse}$ by ROI-Align Ren et al. (2017), which is computed as follows,

$$\mathbf{P}_i^{mlm} = \text{MLP}(\text{ROI-Align}(\mathbf{F}_{fuse}, b_i)), \tag{1}$$

where $b_i$ is the bounding box of the $i$th text region, and the $\mathbf{P}_i^{mlm}$ is optimized by cross-entropy loss with 30,522 token categories.

### 3.2.2 TASK #2: MASKED IMAGE MODELING

In MAE He et al. (2022), BEiT Bao et al. (2022), and SimMIM Xie et al. (2022), patch-level Masked Image Modeling has shown strong potential in representation learning. However, in the document understanding domain, patch-level feature learning is too coarse to represent the details of text or word region. Therefore, we introduce a text region-level visual representation learning task called

Masked Image Modeling to enhance document representation and understanding. Instead of classifying the classes defined by tokenization like LayoutLMv3 and BEiT, we regress the raw pixel values of the masked text region with mean square error loss following SimMIM and MAE. Specifically, we mask the rectangle text regions and predict the RGB values of missing pixels, leading to significant improvement for performance in representation learning.

*Decoder for Task #2.* We develop a Fully Convolutional Network (FCN) with Transpose Convolution to carry out the document image reconstruction of masked text regions. Specifically, we apply the global average pooling to aggregate each text region's feature and generate the embedding $\mathbf{Emb}_{style}$ that mainly represents the visual "style" for each masked text region. To strengthen its text information, we encode the MLM prediction $\mathbf{P}_i^{mlm}$ to $\mathbf{Emb}_{content}$ by using an embedding layer, denoting the "content" knowledge. At last, we concatenate $\mathbf{Emb}_{style}$ and $\mathbf{Emb}_{content}$ and feed it to a FCN, generating the final restored image prediction $\mathbf{P}_i^{mim}$. The procedure of Mask Image Modeling can be formulated as follows,

$$\mathbf{Emb}_{style} = \text{GAP}(\text{ROI-Align}(\mathbf{F}_{fuse}, b_i)), \tag{2}$$

$$\mathbf{Emb}_{content} = \text{EmbeddingLayer}(\mathbf{P}_i^{mlm}), \tag{3}$$

$$\mathbf{P}_i^{mim} = \text{FCN}(\text{Concat}(\mathbf{Emb}_{style}, \mathbf{Emb}_{content})), \tag{4}$$

where GAP is the global average pooling operator. In MIM, we follow MAE and predict the missing pixels of masked text regions. For example, we resize the spatial resolutions of each masked text region to fixed 64×64, and each text-region's regression target is 12,288=64×64×3 pixels of RGB values. The $\mathbf{P}_i^{mim}$ is optimized by MSE loss in the pre-training phrase.

## 3.3 Downstream Tasks

The StrucTexTv2 pre-training scheme contributes to a visual-textual representation with input of image-only. The multi-modal representation is available to model fine-tuning and profits numerous downstream tasks.

### 3.3.1 Task #1: Document Image Classification

Document image classification aims to predict the category of each document image, which is one of the fundamental tasks in office automation. For this task, we downsample the output feature maps of backbone network by four 3×3 convolutional layers with stride 2. Next, the image representation is fed into the final linear layer with softmax to predict the class label.

### 3.3.2 Task #2: Document Layout Analysis

Document layout analysis aims to identify the layout components of document images by object detection. Following DiT, we leverage Cascade R-CNN Cai & Vasconcelos (2018) as the detection framework to perform layout element detection and replace its backbone with StrucTexTv2. Thanks to the multi-scale context design of backbone networks, four resolution-modifying features (P2~P5) of FPN fusion layers on backbone networks are sent into the iterative structure heads of the detector.

### 3.3.3 Task #3 Table Structure Recognition

Table structure recognition aims to recognize the internal structure of a table which is critical for document understanding. Specifically, We employ Cascade R-CNN for cell detection in our table structure recognition framework while replacing the feature encoder with backbone networks. Since some table images are collected by cameras and many cells are deformed, we modify the final output of Cascade R-CNN to the coordinate regression of four vertexes of cells.

### 3.3.4 Task #4: Document OCR

We tend to read the text in an end-to-end manner based on StrucTexTv2. Our OCR method consists of both the word-level text detection and recognition modules. They share the features of backbone networks and are connected by ROI-Align operations. The text detection module adopts the standard DB Liao et al. (2023) algorithm, which predicts the binarization mask for word-level bounding boxes. Similar to NRTR Sheng et al. (2019), the text recognition module is composed of multi-layer Transformer decoders to predict character sequences for each word.

### 3.3.5 TASK #5: END-TO-END INFORMATION EXTRACTION

The aim of the task is to extract entity-level content of key fields from given documents without predefined OCR information. We evaluate the StrucTexTv2 model based on the architecture of document OCR and devise a new branch for semantic entity extraction. Concretely, another DB detection is developed to identify the entity bounding boxes. An additional MLP block is performed with the ROI features to classify entity label. These bounding boxes are utilized for word grouping to merge the text content from Task #4. At length the key information is obtained by grouping words according to the reading order.

## 4 EXPERIMENTS

### 4.1 DATASETS

**Pre-training Data** By following DiT Li et al. (2022), we pretrain StrucTexTv2 on the IIT-CDIP Test Collection 1.0 dataset Lewis et al. (2006), whose 11 million multi-page documents are split into single pages, totally 42 million document images.

**RVL-CDIP** Harley et al. (2015) contains 400,000 grayscale document images in 16 classes, with 25,000 images per class. We adopt RVL-CDIP as the benchmark to carry out experiments on document classification task. Average classification accuracy is used evaluate model performance.

**PubLayNet** Zhong et al. (2019) consists of more than 360,000 paper images built by automatically parsing PubMed XML files. The five typical document layout elements (text, title, list, figure, and table) are annotated with bounding boxes. Mean average precision (mAP) @ intersection over union (IOU) is used as the evaluation metric of document layout analysis.

**WTW** Long et al. (2021) covers unconstrained table in natural scene, requiring table structure recognizer to have both discriminative and generative capabilities. It has a total of 14581 images in a wide range of real business scenarios and the corresponding full annotation (including cell coordinates and row/column information) of tables.

**FUNSD** Jaume et al. (2019) is a form understanding dataset that contains 199 forms, which refers to extract four predefined semantic entities (questions, answers, headers, and other) and their linkings presented in the form. We focus on two tasks of document OCR and end-to-end information extraction on FUNSD. For evaluation, we compute the normalized Levenshtein similarity (1-NED) between the predictions and the ground truth.

### 4.2 IMPLEMENTATION DETAILS

**Pre-training on IIT-CDIP** The proposed encoder network of StrucTexTv2 is composed mainly of the CNN and Transformer. To balance efficiency and effectiveness, StrucTexTv2$_{Small}$ consists of ResNet-50 and 12-layer Transformers (128 hidden size and 8 attention heads) and introduces only 28M parameters. A larger version of StrucTexTv2$_{Large}$ is set up as ResNeXt-101 Xie et al. (2017) and 24-layer Transformers (768 hidden size and 8 attention heads), which total parameters are 238M. We use the networks trained with ImageNet Deng et al. (2009) as the initialization of CNNs. The Transformers are initialized from the language models Sun et al. (2020). StrucTexTv2$_{Small}$ and StrucTexTv2$_{Large}$ take 17 hours and 52 hours to train one epoch of the IIT-CDIP data, respectively. The whole pre-training phase takes nearly a week with 32 Nvidia Tesla 80G A100 GPUs.

**Fine-tuning on RVL-CDIP** We evaluate StrucTexTv2 for document image classification. We fine-tune the model on RVL-CDIP for 20 epochs with cross-entropy loss. The learning rate is set for 3e-4 and the batch size is 28. The input images are resized to 960×960 and maintain its aspect ratio. We use label smoothing=0.1 in the loss function. Besides, the data augmentation methods such as CutMix and MixUp with 0.3 probability are applied on the training phase.

**Fine-tuning on PubLayNet** We evaluate on the validation set of PubLayNet for document layout analysis. We fine-tune the Cascade R-CNN and initialize the backbone with our pre-trained model. The detector is trained 8 epochs with Momentum optimizer and a batch size of 8. The learning rate is set to 1e-2, while it decays to 1e-3 on 3 epoch and decays 1e-4 on 6 epoch. We use random resized cropping to augment the training images while the short edges does not exceed 800 pixels.

Table 1: Performance comparisons on the RVL-CDIP dataset. We report classification accuracy on the test set. $T$ and $I$ denote the text and image modality of input. The proposed StrucTexTv2 achieves a comparable accuracy to the state-of-the-art models with image-only input.

| Methods | Modalities | Accuracy | #Param. |
|---|---|---|---|
| BERT$_{Base}$ *Devlinet al.* (2019) | $T$ | 89.81% | 110M |
| BERT$_{Large}$ *Devlinet al.* (2019) | $T$ | 89.92% | 340M |
| StructuralLM$_{Large}$ *Liet al.* (2021a) | $T$ | 96.08% | 355M |
| SelfDoc Li et al. (2021b) | $V+T$ | 92.81% | - |
| LayoutLM$_{Large}$ *Xuet al.* (2020) | $V+T$ | 94.43% | 390M |
| UDoc Gu et al. (2021) | $V+T$ | 95.05% | 272M |
| TILT$_{Large}$ *Powalskiet al.* (2021) | $V+T$ | 95.52% | 780M |
| LayoutLMv2$_{Large}$ *Xuet al.* (2021) | $V+T$ | 95.64% | 426M |
| LayoutLMv3$_{Large}$ *Huanget al.* (2022) | $V+T$ | 95.93% | 368M |
| **DocFormer**$_{Base}$ *Appalarajuet al.* (2021) | $V+T$ | **96.17%** | 183M |
| BEiT$_{Base}$ *Baoet al.* (2022) | $V$ | 91.09% | 87M |
| DiT$_{Base}$ *Liet al.* (2022) | $V$ | 92.11% | 87M |
| DiT$_{Large}$ *Liet al.* (2022) | $V$ | 92.69% | 304M |
| **StrucTexTv2**$_{Small}$ | $V$ | 93.40% | **28M** |
| StrucTexTv2$_{Large}$ | $V$ | 94.62% | 238M |

**Fine-tuning on WTW** We conduct experiments on WTW for table structure recognition. We also employ Cascade R-CNN to detect the cells of table whose backbone is replaced by pre-trained StrucTexTv2. We fine-tune our model end-to-end using ADAM Kingma & Ba (2015) optimizer for 20 epochs with a batch size of 16 and a learning rate of 1e-4. The input images are resized to $640 \times 640$ after random scaling and the long size being resized to 640 pixels.

**Fine-tuning on FUNSD** On account of the full annotations, both the document OCR and end-to-end information extraction tasks are measured on FUNSD. We set the text recognition network as a 6-layers Transformer and fine-tune the whole model for 1200 epochs with a batch size of 32. We follow a cosine learning rate policy and set the initial learning rate to 5e-4. Extra position embeddings are appended to roi-features and we pass it to each layer of decoders. The training losses except DB detector are the cross-entropy function. Additionally, the same loss is estimated for each decoder layer in the text recognition module for better convergence training.

### 4.3 COMPARISONS WITH THE STATE-OF-THE-ART

To investigate the effect of visual-textual representations, we benchmark StrucTexTv2 with several state-of-the-art techniques on different downstream tasks. Since only small datasets (149 training documents on FUNSD) and to avoid exceeding GPU memory (a tremendous number of table cells on WTW), we only evaluate StrucTexTv2$_{Small}$ on the FUNSD and WTW datasets.

**RVL-CDIP** As the Tab. 1 shows, OCR-based approaches such as DocFormer Appalaraju et al. (2021) with multi-modal modeling takes superior performance on RVL-CDIP. Compared with image-only methods, it relies on stand-alone OCR systems which are low efficiency for practical applications. Our StrucTexTv2$_{Small}$ achieves an accuracy of 93.40% and outperforms previous image-only methods with a much smaller model size. Besides, StrucTexTv2$_{Large}$ drives an accuracy of 94.62%, which brings a 1.93% improvement over DiT$_{Large}$ Li et al. (2022). Intuitively, the results show that StrucTexTv2 is effective for semantic understanding on document image classification.

**PubLayNet** The experiment results on PubLayNet are presented in Tab. 2. It is observed that StrucTexTv2 achieves new state-of-the-art performance of 95.4% and 95.5% mAP scores for both small and large settings. StrucTexTv2$_{Small}$ beats even LayoutLMv3$_{Base}$ Huang et al. (2022) (the result of LayoutLMv3$_{Large}$ is not released in the paper) which contains multi-modal inputs by 0.3%. We suggest that our dual-modal pre-training tasks can learn rich visual-textual representations of document images and performs excellently in confusing situations. Notably, StrucTexTv2$_{Large}$ gets 0.1% mAP

Table 2: Performance comparisons on the PubLayNet validation set. The mAP @ IOU [0.50:0.95] is used as the metric.

| Methods | mAP |
|---|---|
| BEiT$_{Base}$ | 93.1% |
| UDoc | 93.9% |
| DiT$_{Large}$ | 94.9% |
| LayoutLMv3$_{Base}$ | 95.1% |
| StrucTexTv2$_{Small}$ | 95.4% |
| **StrucTexTv2$_{Large}$** | **95.5%** |

Table 3: Performance comparisons on the WTW dataset. The F1-Score is used to measure the accuracy of cell coordinate when IOU=0.9.

| Methods | F1-Score |
|---|---|
| Split+Heuristic Tensmeyer et al. (2019) | 3.4% |
| Faster-RCNN Ren et al. (2017) | 64.4% |
| TridenNet Li et al. (2019) | 65.0% |
| CenterNet Duan et al. (2019) | 73.1% |
| Cycle-CenterNet Long et al. (2021) | 78.3% |
| **StrucTexTv2$_{Small}$** | **78.9%** |

gain over that on StrucTexTv2$_{Small}$. We attribute this subtle improvement to the over-fitting to a certain extent with increasing model size.

**WTW** Tab.3 shows quantitative results of table structure recognition on the WTW dataset. StrucTexTv2 achieves the physical 78.9% F1-score among all published methods. We reconstruct the table structure based on the detection results of table cells. The superior performance of StrucTexTv2 largely due to the proposed pre-training framework.

**FUNSD** We evaluate StrucTexTv2$_{Small}$ on both the document OCR and end-to-end information extraction tasks. As shown in Tab. 4, StrucTexTv2$_{Small}$ achieves outstanding performance, 84.1% 1-NED for document OCR and 55.0% 1-NED for information extraction. Significantly, the whole network is end-to-end trainable. Compared to StrucTexT Li et al. (2021c) and LayoutLMv3 with need of separately stage-wise training strategies, our model alleviates the error propagation in a documental system with key information parsing.

Table 4: Performance comparisons on FUNSD. We present the Normalized Edit Distance (1-NED) for the word-level document OCR and the entity-level information extraction. The * denotes a multi-stage process in which the methods are applied using our OCR results and entity boxes for word grouping in information extraction.

| The results of Document OCR | | The results of Information Extraction | |
|---|---|---|---|
| **Methods** | **1-NED** | **Methods** | **1-NED** |
| DB+NRTR Sheng et al. (2019) | 73.8% | StrucTexT$^*_{Base}$ | 46.8% |
| Google Vision Google (2019) | 76.4% | LayoutLMv3$^*_{Base}$ | 53.1% |
| **StrucTexTv2$_{Small}$** | **84.1%** | **StrucTexTv2$_{Small}$** | **55.0%** |

### 4.4 ABLATION STUDY

To further examine the different contributions of StrucTexTv2, we conduct several ablation experiments, such as document layout analysis on PubLayNet, document image classification on RVL-CDIP, and end-to-end information extraction on FUNSD. All models in ablation study are pre-trained for 1 epoch with only 1M documents sampled from the IIT-CDIP dataset .

Table 5: The ablation study on pre-training tasks and different encoding structures.

| Model | #Param. | MIM | MLM | RVL-CDIP | PubLayNet |
|---|---|---|---|---|---|
| SwinT$_{base}$+FPN | 176M | ✓ | ✓ | 86.6% | 92.2% |
| ViT$_{base}$+FPN | 116M | ✓ | ✓ | 88.6% | 93.2% |
| StrucTexTv2$_{Large}$ | 238M | ✓ | ✓ | **94.1%** | **95.6%** |
| StrucTexTv2$_{Small}$ | 28M | ✓ | | 91.8% | 94.1% |
| StrucTexTv2$_{Small}$ | 28M | | ✓ | 92.0% | 94.5% |
| StrucTexTv2$_{Small}$ | 28M | ✓ | ✓ | **92.5%** | **94.9%** |

**Encoding Structures** In this study, we evaluate the impact of encoding structures by replacing the backbone of StrucTexTv2 with ViT Dosovitskiy et al. (2021) and SwinTransformer Liu et al. (2021). As shown in Tab. 5, the proposed network of StrucTexTv2$_{Small}$ even achieves the better results of 92.5% accuracy and 94.9% mAP on RVL-CDIP and PubLayNet, respectively. The performance of two benchmarks dropped by 3.9% accuracy and 1.7% mAP with the ViT$_{Base}$. Replaced with the SwinTransformer$_{Base}$, the degradation is more obvious. In addition, StrucTexTv2$_{Large}$ improves performance by 1.6% on RVL-CDIP and 0.7% on PubLayNet.

**Pre-training Tasks** In this study, we identify the contributions of different pre-training tasks. As shown in the bottom of Tab. 5, the MIM-only pre-trained model achieves an accuracy of 91.8% on RVL-CDIP and an mAP of 94.1% on PubLayNet. The MLM-only pre-trained model achieves 92.0% and 94.5% for the two datasets. The MLM and MIM can jointly exploit the multi-modal feature representations in StrucTexTv2. By combining both the proposed pre-training tasks, the accuracy is improved to 92.5% on RVL-CDIP and the mAP achieves 94.9% on PubLayNet.

Table 6: The ablation study on the influence of masking ratios (MR.) with StrucTexTv2$_{Small}$ on RVL-CDIP and PubLayNet.

| MR. | RVL-CDIP | PubLayNet |
|-----|----------|-----------|
| 0.15 | 92.1% | 94.7% |
| **0.30** | **92.5%** | **94.9%** |
| 0.45 | 91.7% | 94.8% |
| 0.60 | 92.4% | 94.8% |

Table 7: Consumption analysis on RVL-CDIP. We re-implement LayoutLMv3$^*_{Base}$ with open-source OCR engines to provide text. † denotes the cost of OCR process. All the models are inferred on a NVIDIA Tesla 80G A100.

| Methods | OCR | GPU(MB) | Time(ms) |
|---------|-----|---------|----------|
| LayoutLMv3$^*_{Base}$ | Tesseract | 2,184+n/a† | 22+1,105† |
| LayoutLMv3$^*_{Base}$ | PaddleOCR | 2,184+3,450† | 22+252† |
| StrucTexTv2$_{Small}$ | None | 2,276 | 56 |
| StrucTexTv2$_{Large}$ | None | 4,058 | 284 |

**Masking Ratios** We investigate the effect of training with different masking ratios. As shown in Tab.6, by replacing the masking ratio with 0.15, 0.30, 0.45 and 0.60, the accuracy of RVL-CDIP is 92.1%, 92.5%, 91.7% and 92.4%, respectively. We also report the results on PubLayNet, the mAP of PubLayNet is 94.7%, 94.9%, 94.8%, and 94.8%, respectively. It suggests that the best masking ratio of our pre-training tasks is 0.30. At the same time, It also suggests that the performance of downstream tasks is less sensitive to the selection of masking ratio.

**Consumption Analysis** As shown in Tab. 7, StrucTexTv2$_{Small}$ consumes 56ms and 2,276MB GPU memory to infer one image on RVL-CDIP, while LayoutLMv3$_{Base}$ spends more GPU memory or time with different OCR engines. It is observed that the OCR process of the two-stage method accounts for the vast majority of computation overhead. Thus, our OCR-free framework can achieve a better trade-off between performance and efficiency.

**Masking Strategies** The impact of adjusting text region-level masking to patch-level masking is evaluated in Tab.8. The performance drops a 4.2% accuracy score on RVL-CDIP and a 1.0% mAP on PubLayNet, which demonstrates the effectiveness of the proposed Text-Region masking strategy.

Table 8: Comparison between performance of different masking strategies on RVL-CDIP and Pub-LayNet. The model is only pre-trained with the MIM task.

| Model | Masking strategy | RVL-CDIP | PubLayNet |
|-------|------------------|----------|-----------|
| StrucTexTv2$_{Small}$ | patch-level | 87.4% | 93.1% |
| StrucTexTv2$_{Small}$ | region-level | **91.8%** | **94.1%** |

## 5 CONCLUSION

This work successfully explores a novel pre-training framework named StrucTexTv2 to learn visual-textual representations for document image understanding with image-only input. By performing text region-based image masking, and then predicting both corresponding visual and textual content, the proposed encoder can benefit from large-scale document images efficiently. Extensive experiments on five document understanding tasks demonstrate the superiority of StrucTexTv2 over state-of-the-art methods, especially an improvement in both efficiency and effectiveness.

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
