# OpenReview forum: "StrucTexTv2: Masked Visual-Textual Prediction for Document Image Pre-training"
_ICLR.cc/2023/Conference — ICLR 2023 poster_

### Official Review · Reviewer_oo5U · 2022-10-21

**Confidence:** 4
**Correctness:** 4
**Technical Novelty And Significance:** 3
**Empirical Novelty And Significance:** 3
**Recommendation:** 8

**Clarity, Quality, Novelty And Reproducibility:**

The paper is well written to explain the methods. The idea is intuitive and straightforward and I see a good novelty and value to the document image understanding community. However, the novelty of the work is not very high from a broader AI research perspective and the scope of the impact will be within the particular community.

**Strength And Weaknesses:**

Strengths:

It only takes an image is input, but still provide great results. It effectively gets rid of the need of text input while keeping it in the output. It could be used for almost any tasks for document image understanding.

It less relies on external OCR engines. It could even be used to improve OCR. This is a major difference to other methods.

Weaknesses:

It uses words as units. Both tasks might not be very robust when needing to deal with very short and long words.

From Table 5, it looks like the MaskDoc backbone itself is much superior to other popular backbones. It is not clear if the algorithm is only effective to MaskDoc or the backbone is just very strong.

The last ablation study was conducted with ViT due to some reason. I was not sure why it was not MaskDoc.

**Summary Of The Paper:**

This paper proposes two new pre-training tasks for document image understanding problems such as classification, detection, recognition, and information extraction. The first task, Masked Language Modeling (MLM), to predict the word token for an ROI-Aligned feature map. The second task, Masked Image Modeling, is to reconstruct the pixels corresponding to the word bounding box of concern. Both tasks randomly mask word regions based on the bounding boxes and take only the masked image as input. It enables it to be used for many down streams tasks and reduce the dependency to OCR systems that often required by other relevant methods such as LayoutLMs. The experimental results confirm the effectiveness of the approach.

**Summary Of The Review:**

It presents a concise but effective method to perform pre-training for document image understanding models. There are some concern and the impact could be limited in the domain, but given the practical importance of the domain and the effectiveness of the approach, I'm happy to support acceptance.

---

### Official Review · Reviewer_XqWz · 2022-10-24

**Confidence:** 4
**Correctness:** 3
**Technical Novelty And Significance:** 3
**Empirical Novelty And Significance:** 3
**Recommendation:** 8

**Clarity, Quality, Novelty And Reproducibility:**

The paper is clear and well written and introduces a novel way of pre-training a document representation using only the image as input with OCR as supervision for pre-training. In this sense I consider that the paper has good novelty and contribution. In terms of reproducibility I have some concerns (already commented before) on the OCR used and the strategy to select masked tokens.

**Strength And Weaknesses:**

Strengths
- A novel multimodal pre-training framework for document representation is proposed where only image is used as input. OCR is only used as supervision for the masked language modelling pre-training task, but it is not necessary for inference for downstream tasks. This reduces the dependance on OCR.
- A novel pre-training architecture and pre-training tasks are defined to apply masked language modelling and image reconstruction from masked OCR text regions, with the image reconstruction branch integrating language information coming from the masked language modeling branch.
- The small version of the model has a much smaller number of parameters thant current SoA models while achieving competitive performance (in some cases slightly below the SoA and in other cases slightly above)
- The paper is clear and well written

Weaknesses
- Although the paper claims for avoiding dependance on OCR, OCR is still required for the pre-training (even if only a subset of all OCR tokens in the document are used). In this sense, I miss more details on the OCR and how OCR tokens are selected.  1) Which is the OCR system that has been used in the experiments? Can the framework have some type of dependance on the quality of the OCR used to obtain the tokens? Some analysis on the impact of the quality of the OCR in the final results could be insightful. 2) In the last sentence of section 2, it is stated that "only some high-confidence word-level OCR results need to be randomly selected", but there are no further details on how the selection of OCR tokens is done during pre-training. Are OCR tokens completely randomly selected? Or selection is based somehow on the confidence of the OCR detections? If this was the case, how?
- The large version of the model does not seem to bring about signifficant performance improvement in relation to the large increase in the number of parameters of the model.
- The model has shown pretty good performance on tasks which largely rely on visual information (document classification, layout analysis, table structure recognition) or on semantically simple linguistic information (information extraction, table recognition). I have some doubts on the ability of the model to adapt and perform well on tasks which require a deeper semantic understanding of language, such as DocVQA. Any insight on this?
- I missed some details on the computational resources and time required to train the model. This is an important aspect in these type of models trained on such a large set of data.

**Summary Of The Paper:**

This paper describes a new multimodal framework for pre-training a document image representation that can be used for several downstream document understanding tasks. The main contribution of the proposed framework lies on using only the image as input, while some OCR detections are used only for supervising the language branch in pre-training. This reduces the dependence on OCR of current methods. Specific pre-training tasks are defined for this setting. Experimental results that the proposed framework can obtain results that are comparable or slightly better than current SoA with a much smaller number of parameters.


**Summary Of The Review:**

I consider that the paper is an interesting contribution to the domain of pre-training multimodal document representations. The approach is novel and results are good enough. As commented before, there are a few issues that could be better discussed and analysed in the paper, mainly the role of the OCR and the application to more language-oriented tasks, but overall I think that the paper has more positive contributions than weaknesses.

---

### Official Review · Reviewer_vxyY · 2022-10-25

**Confidence:** 4
**Correctness:** 3
**Technical Novelty And Significance:** 4
**Empirical Novelty And Significance:** 3
**Recommendation:** 6

**Clarity, Quality, Novelty And Reproducibility:**

* The paper is quite clear, well-illustrated and well-written (although, there are some mistakes to fix).
* The model is described in great details, and I feel like this work could be easily reproduced. Beside, the code and model will be released.

**Strength And Weaknesses:**

##### Strong points
* The experiments are convincing. The model obtains strong/SOTA results for 5 tasks related to Document Image Understanding.
* The pretraining framework strategy is an interesting contribution that is relevant to ICLR, and could be useful for other researchers.
* The ablation study is very interesting, especially the study of the masking ratio, masking strategies and pretraining tasks.

#### Weak points

* My main concern is that I am not convinced that using visual-textual representations (MIM+MLM) is really that helpful. At least for Document Layout Analysis, pretraining on both tasks imporve performance by less than 1% (in Table 5). It would be **very** interesting to complete this ablation study on other tasks (especially on tasks that are more text-dependent than DLA). I feel like this is the heart of the article: is visual-textual pretraining actually better than visual-only or textual-only pretraining ? This question is not fully answered in the paper.
* It's not clear to me how the text is obtained/tokenized for each text zone. I understand that an OCR is used, and random words with high confidence scores are selected to be masked. However, how is the sub-word tokenization performed ? And why the authors only keep the first sub-word during pretraining ("we only choose the first sub-word for keeping the number of words in the document images") ? I feel like this step is crucial for reproducibility, and should be explained more clearly.
* The authors claim that using an OCR is 1) costly 2) sub-optimal as image/text features are not jointly optimized. These claims are not really supported in the paper. For 1) it would be intersting to compare MaskDoc to LayoutLM-v3 or DocFormer in terms of computing time/resources. For 2), it makes sense, but the classification results (Table 1) show that LayoutLM-v3 and DocFormer outperform MaskDoc. It would be interesting to comment these results in Section 4.3.
* The ablation study comparing different masking strategies is not sufficiently detailed, especially since the authors insist that masking text zones is better than masking patches.
* The use of CutMix and MixUp data augmentation on RVL-CDIP is not detailed. Do the other models use this data augmentation strategy as well ? What is the impact on performance ?
* There is no information regarding the computing requirements, memory consumption and training/inference time for MaskDoc.

#### Minor comments:
* Some typos/english mistakes should be fixed:
    * abstract "comprises ~~of~~ two self-supervised" (or "consists of")
    * 3.1 a backbone network "consist**s**
    * Fig2. "to reconstruct~~s~~"
    * 3.2.1 "does not need **to** convert"
    * 3.3.1 "a**n** embedding layer"
    * 3.3 "to strength**en**"
    * 4.2 "The ~~rest~~ other parameters"
    * 4.4 "pretrain**ed** models"
    * ...
* Appearing twice in the references:
    * Adam W Harley, Alex Ufkes, and Konstantinos G Derpanis. Evaluation of deep convolutional nets for document image classification and retrieval. In ICDAR, pp. 991–995, 2015b
    * Kaiming He, Xinlei Chen, Saining Xie, Yanghao Li, Piotr Doll ́ar, and Ross Girshick. Masked autoencoders are scalable vision learners. CVPR, 2022b.
* 8 articles are cited as arxiv preprints - please update the citation for articles that have been published.


**Summary Of The Paper:**

The paper introduces Mask-Doc, a new self-supervised pre-training framework for automatic Document Image Understanding that learns from a single input modality (image). Two pretraining tasks (*Masked Image Modeling* and *Masked Language Modeling*) are learned in an end-to-end manner to extract visual-textual representations.

First, an off-the-shelf OCR is used to mask 30% of text zones in a document image.
The masked image is passed through a feature extractor composed of CNNs and Transformer. Features are merged and ROIAlign is applied to extract the features of each masked zone.
Finally, two sub-networks are trained to:
1) reconstruct the pixel values of the masked zone ;
2) reconstruct the textual content corresponding to the masked zone.

This architecture is pre-trained on the IIT-CDIP Test Collection 1.0 dataset (large-scale document dataset) and fine-tuned on 5 downstream tasks related to document image understanding: Document Classification (on RVL-CDIP), Document Layout Analysis (on PubLayNet), Table Structure Recognition (on WTW), Optical Character Recognition (on FUNSD), Information Extration (on FUNSD). MaskDoc obtains strong results on every task and SOTA results on four of them.


**Summary Of The Review:**

The proposal to carry out a combined pre-training on both images and text is interesting and well evaluated. It would be interesting to demonstrate in which type of application the model actually outperforms its competitors. More technical details should be provided on some aspects.

---

### Official Review · Reviewer_HUEm · 2022-10-26

**Confidence:** 5
**Correctness:** 3
**Technical Novelty And Significance:** 2
**Empirical Novelty And Significance:** 2
**Recommendation:** 5

**Clarity, Quality, Novelty And Reproducibility:**

* The novelty is needed to be justified as the motivation is not clear and the discussion of missing related works.
* The code is not released, so I'm not sure about the reproducibility.

**Strength And Weaknesses:**

* The major question I have is the motivation of the work.  It is not clear to me why we would like to discard OCR as it is quite matured and well-developed, also it can leverage external data to train the OCR model.  The argument of the cost of running OCR does not convince me either.  The QPS of the modern OCR system is quite high (e.g. on major cloud platforms). Therefore, it seems not well-motivated to me.  Regardless of using OCR in the final algorithm,  the possibility can be on improving the image component by absorbing what OCR is doing, then showing how it can improve the final performance together with text components with OCR. If the authors provide some experiments on this, it would be more promising.
* Missing important related work, “OCR-free Document Understanding Transformer”, which is published in ECCV 2022. How does the proposed work compare with this earlier work?  A separate note: although this work also shares the same motivation of being OCR free, it doesn’t really justify why we want an algorithm that discards OCR as it is quite standard in the representative works and it is actually cheap in practice.
* It is surprising that the small model can achieve almost on-par performance with large models in Table 1. However, it looks a bit too good to be true for complex language understanding tasks. So does it mean the classification is actually a relatively simple task? If so, I would recommend the authors to try on more challenging NER tasks, as LayoutLMs and FormNet (Lee et al., 2022) have studied to demonstrate how much the proposed algorithm really learn the task semantics.


**Summary Of The Paper:**

The authors propose a new visual-text learning algorithm for document understanding. In particular, the authors argue that the advantage of proposed algorithm is free of using OCR in the prediction.

**Summary Of The Review:**

The novelty is needed to be justified as the motivation is not clear and the discussion of missing related works.  Also, it would be nice to showcase on more challenging NER tasks.

---

### Decision · Program_Chairs · 2023-01-20

**Decision:**

Accept: poster

**Justification For Why Not Higher Score:**

The paper has good motivation and experimental results. However, the technical contributions of the paper are quite limited.

**Justification For Why Not Lower Score:**

N/A

**Metareview: Summary, Strengths And Weaknesses:**

This paper introduces a novel pretraining framework named MaskDoc to learn visual-textual representations for document image understanding with a single image modality input. In general, the paper is well-written and easy to follow. The reviewers agreed that this is a good paper with interesting motivation and convincing experiments. The reviewers raised some questions about the motivation and experimental analysis, and the authors addressed them well in the response.

**Note From Pc:**

if the above contains the word "oral" or "spotlight" please see: "oral" presentation means -> notable-top-5% and "spotlight" means -> notable-top-25%. As stated in our emails, we are disassociating presentation type from AC recommendations